# Validation of HIV-1 MA Shell Structural Arrangements and Env Protein Interactions Predict a Role of the MA Shell in Viral Maturation

**DOI:** 10.3390/v15040893

**Published:** 2023-03-30

**Authors:** Tarana A. Mangukia, Joy Ramielle L. Santos, Weijie Sun, Dominik Cesarz, Carlos D. Ortíz Hidalgo, Marcelo Marcet-Palacios

**Affiliations:** 1Department of Medicine, Alberta Respiratory Centre, University of Alberta, Edmonton, AB T6G 2S2, Canada; 2Faculty of Biology, University of Havana, Havana 10400, Cuba; 3Department of Biological Sciences Technology, Laboratory Research and Biotechnology, Northern Alberta Institute of Technology, Edmonton, AB T5G 2R1, Canada

**Keywords:** HIV-1, matrix shell, hexameric matrix trimers, HIV-1 structure, Envelope Cytoplasmic Domain, Env CT

## Abstract

The molecular structure of the type 1 human immunodeficiency virus (HIV-1) is tightly linked to the mechanism of viral entry. The spike envelope (Env) glycoproteins and their interaction with the underlying matrix (MA) shell have emerged as key components of the entry mechanism. Microscopy evidence suggests that the MA shell does not span the entire inner lipid surface of the virus, producing a region of the virus that completely lacks an MA shell. Interestingly, evidence also suggests that Env proteins cluster during viral maturation and, thus, it is likely that this event takes place in the region of the virus that lacks an MA shell. We have previously called this part of the virus a fusion hub to highlight its importance during viral entry. While the structure of the MA shell is in contention due to the unaddressed inconsistencies between its reported hexagonal arrangement and the physical plausibility of such a structure, it is possible that a limited number of MA hexagons could form. In this study, we measured the size of the fusion hub by analysing the cryo-EM maps of eight HIV-1 particles and measured the size of the MA shell gap to be 66.3 nm ± 15.0 nm. We also validated the feasibility of the hexagonal MA shell arrangement in six reported structures and determined the plausible components of these structures that do not violate geometrical limitations. We also examined the cytosolic domain of Env proteins and discovered a possible interaction between adjacent Env proteins that could explain the stability of cluster formation. We present an updated HIV-1 model and postulate novel roles of the MA shell and Env structure.

## 1. Introduction

Human immunodeficiency virus-1 (HIV-1) attacks the immune system through efficient viral entry and infection. The virus’ function and mechanism of viral entry are highly tied to its structural components, particularly those located in the periphery of the virus. Creating highly accurate models of HIV-1 particles is, therefore, necessary to understand the mechanisms of viral entry. The structure of HIV-1 particles includes an outer phospholipid bilayer with numerous embedded viral spike envelope (Env) glycoproteins. These proteins originate from the larger glycoprotein-160 (gp160) precursor which is cleaved by proteases to produce glycoprotein-120 (gp120) trimers. An average of 14 Env proteins are found clustered together in locations where there are gaps in the MA shell as the large cytoplasmic (CT) domain of Env proteins requires a significant space on the same plane as the MA shell [1]. Env proteins have been implicated in the receptor-mediated fusion of the viral HIV-1 particle to a host cell’s outer membrane. Multiple Env proteins are potentially involved, suggesting that the clustering of these proteins is required for viral entry [2,3]. X-ray crystallography has determined that MA proteins assembled into a trimeric form [4]. These trimers originate from a group specific antigen (Gag) polyprotein which may be recruited to the inner bilayer. Though the greater superstructure of the MA shell has not been elucidated, current evidence from computational, microscopic, and biomolecular techniques has established that MA trimers assemble into a hexagonal matrix when aggregated on flat, two-dimensional surfaces [1].

Some reported models of the MA shell extrapolated the two-dimensional hexagonal matrix formed by MA trimers and applied it similarly to three-dimensional models. This results in a spherical or ellipsoidal structure of hexagonally arranged MA trimers with each hexagon representing six MA trimers. However, this creates mathematical unfeasibility. A hexagonal matrix arranged on a two-dimensional surface consists of regular hexagons with adjoining angles of 120°. However, upon converting this into a three-dimensional structure, a curvature is introduced. This results in gaps at the conjoining point of hexagons while reducing external angles to less than 120° (Appendix A [5]). Subsequent adjacent additions of hexagons will, therefore, not be able to accommodate the 120° angle required for regular arrays, resulting in overlapping hexagons [5,6]. In a molecularly accurate model of the HIV-1 MA shell, this translates to trimers with overlapping protein volumes in a thermodynamically impossible configuration. Therefore, such models harbour numerous geometric errors, including overlapping MA trimers and gaps within the overall shell, creating an amorphous array of MA trimers.

Upon the discovery and careful assessment of the geometric restrictions of previously published MA shell models, we proposed an alternative MA shell model consisting of a six-lune hosohedron complete with Env glycoproteins embedded within the shell [6]. Our model was able to accommodate both spherical and ellipsoidal structures without creating trimer overlap. We achieved this by creating a three-dimensional structure in which MA trimers were not arranged in hexagons, but rather in incomplete hexagons and lines. This provided a mathematically feasible model of MA shell trimers without creating large gaps within the shell. We further updated the model to include the random incorporation of Env proteins which required a complete resolution of the full structure of the Env protein complex, including the modelling the Kennedy sequence (KS) and CT domain of the Env protein. Env proteins were ultimately found to be situated in gaps between lunes of the hosohedron, dubbed “interlunar gaps”, and can move within them, demonstrating the ability of Env proteins to easily cluster during viral entry. Overall, our model was able to resolve several microscopy observations, such as the diverse nature of the three-dimensional viral structure [7] and the clustering of Env proteins during maturation and viral entry [2,3].

Previous publications by our lab regarding the six-lune hosohedron structure did not address certain critical aspects of the structure of the HIV-1 virus, such as the inclusion of an MA shell fusion hub gap [5], which we improve upon now. Cryogenic electron microscopy (cryo-EM) maps from immature HIV particles have identified potential gaps in the MA shell. These gaps likely represent the developing fusion hub and, therefore, need to be included in our model of the MA shell for a more accurate depiction. We also seek to understand and model the potential molecular interactions between Env proteins which could contribute to the overall clustering phenomenon. In this paper, we improve upon our previously proposed model by incorporating the average size of HIV-1 virus particles, proposing several novel interactions between the CT domain of Env glycoproteins, and the inclusion of an MA shell gap. We carefully validate previously published hexagonal MA shells to assemble a structurally complete and molecularly accurate viral structural model of the HIV-1 MA shell and Env proteins.

## 2. Materials and Methods

### 2.1. Quantification of MA Shell Diameter and Gap

Cryogenic electron microscopy (cryo-EM) maps of HIV-1 particles, consisting of the phospholipid bilayer and matrix (MA) shells, were compiled from EMD-4020 and EMD-13085 from the electron microscopy database (https://www.ebi.ac.uk/emdb/, accessed 9 November 2021). Maps were imported into a molecular visualization and analysis system, PyMOL (Molecular Graphics System, Version 1.7.6.0, Schrödinger, LLC, New York, NY 10036, US), and optimized for viewing using color and saturation manipulation. Particles were chosen for measurement based on our ability to visualize the diameter and MA shell fusion hub gap. Pseudoatoms were manually added at the edges of the fusion hubs. Corresponding distances between these particles were then used to assess the diameter and gap of the MA shell. The manual placement of pseudoatoms was conducted twice and independently by three investigators. Statistical analysis was then performed to determine the average diameter of the MA shell and fusion hub gap.

### 2.2. Visualization of HIV-1 MA Shells from the Literature

Models of HIV-1 MA shells were compiled from the recent scientific literature [8,9,10,11]. Each model was converted into a simple black and white tile model depicting hexamers and their constituent trimers. We developed a software to facilitate this process and it is now currently available at our site below. The software provides a flat lattice of hexagons for a user to manually click potential MA hexagons. The software then exports trimer coordinates that can then be opened in a 3D visualization software such as PyMol or Chimera. The pipeline is depicted in Figure 1. Using PyMOL, a molecularly accurate hexagonal HIV-1 matrix shell of MA trimers was used to then model the HIV-1 MA shells gathered from literature. Rather than employ the six-lune hosohedron structure that our lab previously proposed, we arrayed our MA trimers in a hexagonal array, replicating the proposed structure of other research works from the academic literature. This was carried out to elucidate whether models proposed in literature do in fact result in trimer overlap. All scripts and calculations for this model were developed previously within our lab. Coordinates were exported to PyMol for the construction of structures. All scripts and calculations are available for download (https://sites.ualberta.ca/~marcelo/HIV-1_MA_Builder, accessed on 25 July 2021). Each completed model and its constituent trimers were assessed for overlap with neighboring trimers and color coded to their degree of overlap.

### 2.3. Interaction Models for Two Env CT Domains

The cytoplasmic tail protein structure was obtained from the Protein Data Bank with the PDB ID 5VWL in the PDB format [12]. The ClusPro (https://cluspro.org, accessed 5 May 2022) [13,14,15] and HDOCK (http://hdock.phys.hust.edu.cn, accessed 4 May 2022) [16,17,18,19] online servers were applied to perform protein–protein docking for two cytoplasmic tail proteins. For docking with the ClusPro server, the Multimer Docking option (Beta version) was selected because the same molecule was used as both receptor and ligand [15]; only the structure of the receptor and the number of subunits, two in this case, were indicated. No chains were specified as the structure contains only one chain. The top-scoring models for the four different energy parameters (Balanced, Electrostatic, Hydrophobic and Electrostatic + van der Waals) were downloaded and visualized with PyMOL (Molecular Graphics System, Version 1.7.6.0, Schrödinger, New York, NY 10036, US). For docking with the HDOCK server, a procedure similar to that explained above was followed. The same molecule was used as receptor and ligand (5VWL). No advanced option was selected; therefore, the docking process was based on a hybrid algorithm of template-based modeling and ab initio free docking. The best predictions were downloaded and visualized with PyMOL.

## 3. Results

### 3.1. Estimation of MA Shell Diameter and Fusion Hub Gap Dimensions

Eight immature HIV particles were analyzed in PyMOL to obtain volume density information and trace the extent of the MA shell in 3D (Figure 2); see Materials and Methods. To enhance the interpretation of the maps, we color-coded the various compartments yellow for the lipid bilayer, green for the MA shell layer within the plane of the measurement, and blue for the MA layer in the background of the volume. Our results indicate the average MA shell diameter was assessed as 99.437 nm while the MA shell fusion hub gap was 66.337 nm ± 15.0 nm (Figure 3A). Based on our results, the fusion hub is approximately 2/3 the size of the diameter (Figure 3C). A schematic representation of the model is shown in Figure 3C.

### 3.2. Validation of Published MA Shell Models

We conducted a literature search to locate published peer-reviewed articles that had used schematic representations of HIV-1 MA shells using hexagons or triangles. We located four papers and a total of six images were used as raw data for our validation [8,9,10,11]. These conceptual images were first recreated using our in-house software that takes a 2D hexagon lattice and exports the coordinates of the hexagons into our previous MA builder software [6]. The latter can then be used to reconstruct 3D atomically accurate MA trimers accurately representing structures reported in the literature. Once we created atomic resolution models, we color-coded the trimers according to their proximity and overlap to adjacent trimers. Trimers that had zero structural overlap with any adjacent trimer and had a center-of-mass to center-of-mass distance of >5.2 nm were considered plausible locations and were colored green. Trimers with zero overlap with adjacent trimers with a distance of <5.2 nm between center-of-mass to center-of-mass of adjacent trimers were colored yellow. Orange trimers had approximately 5% structural overlap with adjacent trimers and trimers with higher than 5% overlap were colored red.

In all the generated models, a similar pattern emerged in that a maximum of 4 adjacent MA hexagons were tolerated (Figure 4C) in the curvature of an MA shell of 100 nm diameter. Departing away from these four-by-four initial MA hexagons, six arms of adjacent hexagons radiate out. This is not surprising as single adjacent hexagons can curve to form cylindrical structures as we have previously reported [5,6]. However, a large percentage of the reported MA hexagons were colored yellow, orange, or red, indicating a significant degree of error in the postulated structures.

### 3.3. Interactions of Two Env CT Domains

The CT domain of Env proteins is uncharacteristically large and contains amphiphilic residues that allow the CT domain to interact with the inner membrane. We had hypothesized that this particular property could also allow the interaction of two adjacent CT domains allowing polar residues to interact with each other resulting in hydrophobic residues pointing outward thus stabilizing interactions with the lipid bilayer. We performed a preliminary APBS electrostatic analysis in PyMOL and discovered that such interactions could be compatible with the amino acid sequences of the CT domain (Figure 5A). To validate this hypothesis, we performed a CT-to-CT interaction analysis using two software, HDOCK and ClusPro (Figure 5B,C). These protein–protein docking predicted interactions explain Env protein clustering and predict a mechanism of stabilization of the interaction. Figure 5D shows a depiction of three adjacent Env proteins interlocked via interactions between their CT domains.

### 3.4. An Updated Model of the HIV-1 MA Shell

Careful analysis of all MA shell models postulated in the literature led us to a model that accommodates MA shell trimers in hexagonal configurations. It must be noted that for this model to be possible, none of the hexagons are regular hexagons. This means that one or more trimer-to-trimer distances are compressed to less than 5.2 nm. We added this degree of tolerance to understand the patterns that could potentially emerge and discovered that a cluster of 3 by 3 MA hexagons is the maximum number tolerated. Beyond these nine hexagons, six single, adjacent hexagon rows radiated around the particle (Figure 6A). To account for the fusion hub we measured, we encrypted the extent of the MA shell to retain a gap of 63.2 nm. A total of 14 Env proteins were placed randomly in the MA shell structure as previously carried out [5], allowing 6 Env proteins to interact at the fusion hub (Figure 6B,C).

## 4. Discussion

In an attempt to determine the structural presence of MA shell hexagons, we performed an extensive search in all public cryo-EM databases. In this search, we could not identify any evidence of hexagon formation and, thus, were not able to confirm their presence through direct evidence. In the map files we identified, we were able to study the electron density volume maps and were able to access the extent of the location of the MA shell within viral particles (Figure 2 and Figure 3). We discovered that the MA shell appeared to be continuous, depending on the angle of the examined plane, but also exhibited gaps. This was in full agreement with the six-lune structure we have previously shown [5,6]. Interestingly, we observed that all viral particles had a region of 63.2 nm that did not contain MA trimers. This is consistent with our current model of the MA shell, and it is likely a functional gap that facilitates Env protein dimerization and clustering.

A recent paper by Kun Qu et al. [8] showed cryo-EM tomography images of three adjacent hexagons. To our knowledge, this work is the first published EM electron density map that shows direct evidence of hexagon-to-hexagon interactions. This evidence led us to update our MA shell model to include hexagons. Unfortunately, the map files were not available and only the images shown within the paper can be analyzed. Thus, we carefully extracted the triangle tile data from this manuscript and included this MA shell in our analysis.

Due to the lack of published cryo-EM maps, we adopted a strategy that used the schematic models from peer-reviewed journals. We identified four publications containing a number of conventional tile models [8,9,10,11]. A total of six models were built, using either triangles or hexagons tiles to represent MA hexagons. We then developed an in-house software that would allow us to easily reconstruct the published MA arrangements into a 2D lattice (Figure 1, middle image). The software then exported the 3D coordinates of each node. With the coordinates available, we could then use our MA builder software [5,6] to export the atomic resolution 3D model. At this point, we were able to measure the center of mass of each MA trimer, calculate the distances, and visually access any overlap. Because it is impossible to generate a sphere with regular hexagons, we measured the possibility that in some cases, under some thermodynamic constraint, a model could be built by “pressing” adjacent trimers closer together than distances measured experimentally (Appendix A). Thus, we developed a color code to highlight this accommodation and recreated MA shell models in 3D (Figure 4). We found that the models we studied were not possible geometrically and strongly recommend that validations as outlined in this manuscript are used prior to future publication. The software we developed is free for academic purposes and should be used to replace triangle or hexagon-based models that can be misleading and difficult to interpret by the viewers.

We had previously hypothesized that the CT domains of Env proteins are likely to interact due to their matching amphiphilic residue patterns (Figure 5A). To test this hypothesis, we performed protein docking in silico analysis and found using HDOCK and ClusPro a number of docking structures that agreed with our initial hypothesis (Figure 5). We modeled these interactions using a full-atom-scale Env protein to depict how this interaction could take place. We then generated an MA shell-Env protein model (Figure 6) to summarize our observations. The model contains a 3 by 3 hexagonal arrangement from which six lines of hexagonally arranged MA trimers radiate around the particle sphere. These long chains of consecutive hexagons are not a surprising finding given that this arrangement is similar to what would be observed in cylindrical MA structures (Appendix A) and what was observed in tubular arrays by Bharat et al. [20]. Interestingly, the 3 by 3 hexagonal arrangement would require energy to compact adjacent MA trimers closer than 5.2 nm. This hexagon arrangement may play a key role during viral entry by storing energy that could be released to promote the fusion and transfer of the HIV core. The flat configuration of the 3 by 3 hexagons is the relaxed, more stable arrangement. Bending such a structure into the curvature of a sphere during viral budding could, therefore, be a novel mechanism in viral assembly to store energy in the MA shell that can later be harnessed to eject the core and return the MA shell to a flat configuration.

The interaction between Env and the MA shell has been a focus of other researchers. Prasad et al. [21] used a cryo-electron tomography of HIV-1 virus-like particles (VLPs) that display elevated levels of Env trimers on their surface. This is likely feasible due to a significant 102 amino acid deletion of the C-terminal CT domain of the gp41 protein. It is, thus, not surprizing that the truncated gp41 can adopt 3D shapes that accommodate the Env trimer above the plane of the MA shell, as discovered in their electron tomography. We have previously described that the size of a full CT domain is predicted to occupy a 3D volume similar to that of an MA monomer [5]. Thus, it is unlikely that the interactions between the MA shell and Env trimers described by Prasad et al. occur in WT HIV-1 particles. The limitations resulting from CT domain deletions are also observed in the NMR structures determined by Piai et al. (PDB IDs: 6UJU and 6UJV) [22]. In their study, the deletion of only 22 amino acids in the CT domain resulted in a plate-like arrangement of the remaining structure similar to that reported by Prasad et al. As a result, we were unable to use the NMR structures reported by Piai et al. or the cryo-electron tomograms by Prasad et al.

In this study, we update the structural model of the HIV-1 MA shell and make software available to the community to validate their findings. We recommend that all cryo-EM maps should be made public to help the community analyze maps in silico. The hexagon-based ejection of the HIV core, the interaction of Env proteins via their CT domains, and the clustering of Env proteins in the fusion hub are interesting predictions that will help us further understand the biology of HIV-1.

## 5. Conclusions

In summary, we combined the knowledge accumulated on the structure of the MA shell and the Env trimers to create an updated model of the HIV-1 particle. We developed and made publicly available the software needed to build and test MA shell models in the hope that the community will avoid the use of hexagon tiles in their models. The size of the MA shell gap in which Env proteins likely interact was determined and, thus, we postulate that this region of the virus facilitates interactions between Env proteins via their CT domain. The bioinformatics software we developed in this study will help us validate future cryo-EM data maps, a necessary step needed in data validation.

## Figures and Tables

**Figure 1 viruses-15-00893-f001:**
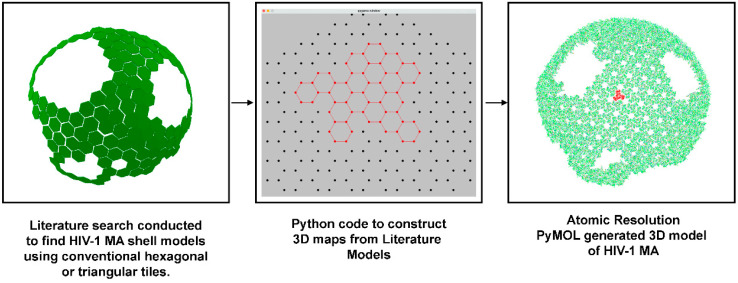
HIV-1 Model Pipeline. HIV-1 MA Shell Models from Literature. A recent literature search was conducted to find conventional HIV-1 shell models [8,9,10,11]. Each green hexagon represents six trimers. Python code was used to generate 3D maps from selected literature models. PyMOL was used to generate molecularly accurate 3D hexagonal models of the HIV-1 MA shells and their trimers at atomic resolution. A trimer is highlighted in red.

**Figure 2 viruses-15-00893-f002:**
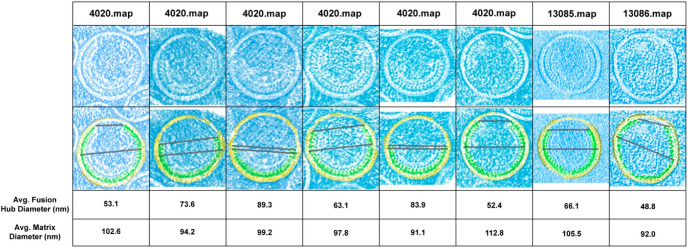
Cryo-EM HIV-1 particles with measurements for the diameter and gap of the matrix (MA) shell. Top row represents raw data and the corresponding map number. Second row shows areas highlighted in green that represent the HIV-1 MA shell. Blue areas represent the remaining HIV-1 particle components and background signal. Yellow areas represent the lipid layer. Particles are overlaid with measurements of the HIV-1 MA shell diameter and gap.

**Figure 3 viruses-15-00893-f003:**
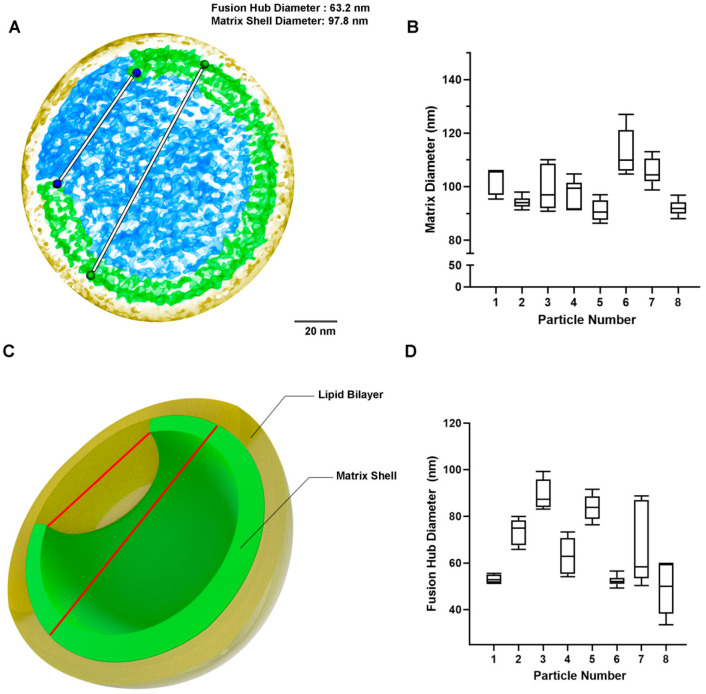
Visualization and measurements of the HIV-1 MA shell diameter and fusion hub gap. (**A**) A representative Cryo-EM particle from EMD-4020 is shown with distances between same-colored pseudoatoms representing the matrix shell diameter and fusion gap length. The distance between the two blue pseudoatoms is the length of the fusion hub diameter, while the distance between the two green pseudoatoms is the length of the fusion hub diameter. (**B**) HIV particle fusion hub measurements. A total of 8 HIV-1 particles were assessed from EMD-4020 and EMD-13085. (**C**) A 3D conceptual image of an HIV-1 particle. The outer yellow color denotes the phospholipid bilayer, whereas the inner green color denotes the matrix shell. The fusion hub gap can be seen by space marked with the shorter red line, marking the fusion hub diameter. The longer red line marks the matrix shell diameter. (**D**) HIV particle matrix diameter measurements.

**Figure 4 viruses-15-00893-f004:**
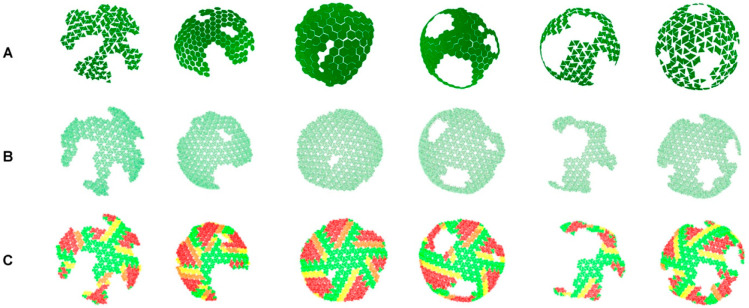
Models of the HIV-1 Shells from Literature (Colored Trimer Model). (**A**) Schematic 3D models based on 2D figures of published MA shell arrangements. Each hexagon representing six MA trimers. Data from (**A**) were inputted into the pipeline shown in Figure 1, and the output is shown in (**B**). Each model and its constituent trimers were assessed for overlap with neighboring trimers and color-coded to their degree of overlap (**C**). Trimers that had zero structural overlap with any adjacent trimer and had a center-of-mass to center-of-mass distance of 5.0-5.2 nm were considered plausible locations and were colored green. Trimers with zero overlap with adjacent trimers with a distance of 5.0–4.4 nm were colored yellow. Orange trimers had approximately 5% structural overlap with adjacent trimers (4.4–4.0 nm) and trimers with higher than 5% overlap were colored red.

**Figure 5 viruses-15-00893-f005:**
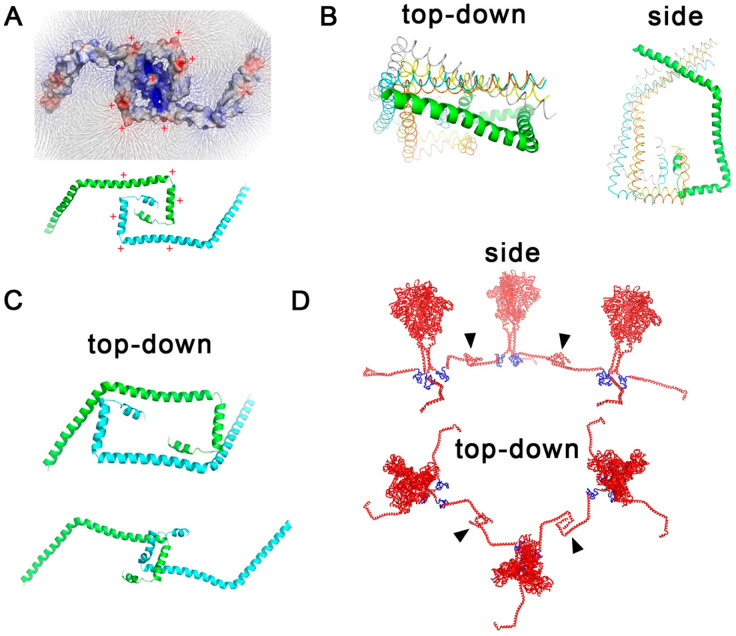
Interaction models of two or more Env CT domains from adjacent Env proteins. (**A**) APBS electrostatic analysis model. Positive residues are colored red and highlighted with positive signs. Blue residues are negatively charged. (**B**) Model showing the best four solutions determined with HDOCK. A “top-down” view, as would be seen from outside of a virion, and a “side” view, as would be seen parallel to the membrane, are shown. (**C**) Models showing the best two solutions using ClusPro. Top-down views are shown. (**D**) Side and top-down views of three Env protein complexes interacting. Arrow heads show interacting segments within the CT domain.

**Figure 6 viruses-15-00893-f006:**
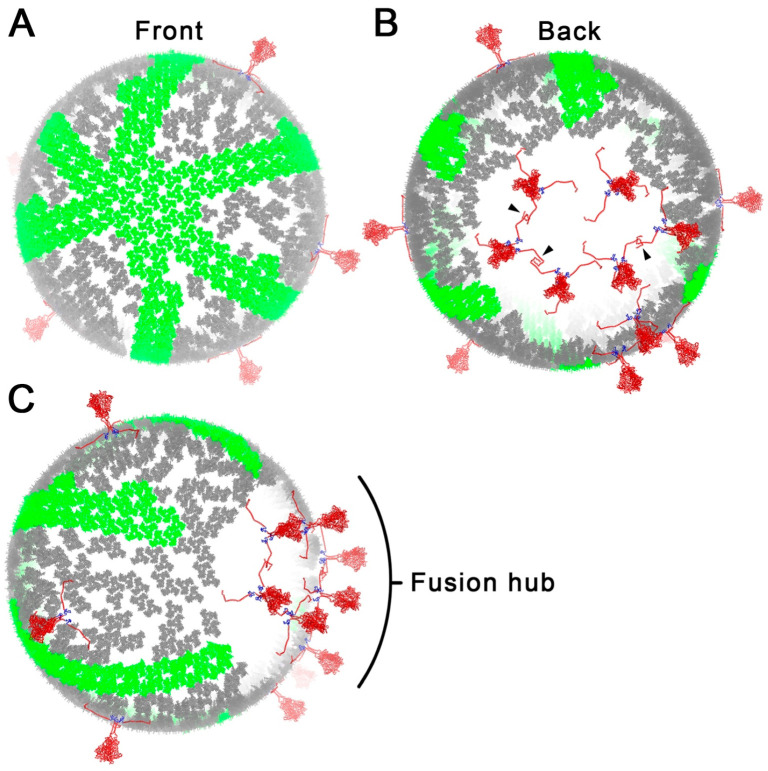
Conceptual model containing hexagonally arranged MA trimers and clustering of Env proteins prior to viral entry. Mathematically feasible trimers are shown in green while the Env protein is depicted in red and blue. (**A**) Front view of a hypothetical HIV-1 particle with mathematically feasible trimers in green and Env proteins in red scattered throughout the matrix. Grey MA trimers do not form hexagons. (**B**) Back view of (**C**) showing the fusion hub opening where Env proteins interact. Arrow heads point to the interactions between adjacent Env proteins promoting clustering. (**C**) Perspective view showing the fusion gap.

## Data Availability

Not applicable.

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
