# Peer review of "Validation of HIV-1 MA Shell Structural Arrangements and Env Protein Interactions Predict a Role of the MA Shell in Viral Maturation"

_viruses, 2023, doi:10.3390/v15040893_

Round 1
Reviewer 1 Report
In this manuscript, an updated HIV-1 model and novel roles of the MA shell and Env structure are presented. This study is original that includes update the structural model of the HIV-1 MA shell and makes software available to the community to validate their findings. Authors have studies in this area and they move forward their research in this manuscript. They used methodology extensively. The references are appropriate but they can expand this part with more references. The quality of figures are very good thanks to the programmes that they chose. The conclusion part is not mandatory but they can add this part to sum up the whole study. This study can be accepted in “viruses”.
Author Response
Reviewer 1
Comments and Suggestions for Authors
In this manuscript, an updated HIV-1 model and novel roles of the MA shell and Env structure are presented. This study is original that includes update the structural model of the HIV-1 MA shell and makes software available to the community to validate their findings. Authors have studies in this area and they move forward their research in this manuscript. They used methodology extensively. The references are appropriate but they can expand this part with more references. The quality of figures are very good thanks to the programmes that they chose.
We thank reviewer 1 for their supportive comments.
The conclusion part is not mandatory but they can add this part to sum up the whole study. This study can be accepted in “viruses”.
We have added a conclusion paragraph to the manuscript.
In summary, we combine the knowledge accumulated on the structure of the MA sell and the Env trimers to create an updated model of the HIV-1 particle. We developed and made publicly available the software needed to build and test MA shell models in hopes that the community will avoid the use of hexagon tiles in their models. The size of the MA shell gap in which Env proteins likely interact was determined and thus we postulate that this region of the virus facilitates interactions between Env proteins via their CT domain. The bioinformatics software we have developed in this study will help us validate future cryo-EM data maps, a necessary step needed in data validation.
Reviewer 2 Report
Despite the human immunodeficiency virus type 1 (HIV-1) was discovered in the early 1980s, significant knowledge gaps remain about the superstructure of the HIV-1 matrix (MA) shell. Understanding the HIV-1 structure and the interaction with Env glycoproteins is a key step for the HIV-1 entry process. The authors have already reported that Env proteins clustered in a space without MA shell, named “fusion hub”. In addition, there are different models of HIV-1 MA described in recent scientific literature, but in the present study, the authors propose an improved HIV-1 structural model. MA trimers are arranged in incomplete hexagons and lines, and Env proteins are packaged in the gap, which measures 66.3 nm ± 15.0 nm.
An extensive editing of English language and style are required (line 32, “Type 1 human immunodeficiency virus” is incorrect; line 244, “we did identify” is incorrect).
Please, revise the reference style. It is not that recommended by the journal guidelines. Line 52, reference is missing; line 237, “HIV paper 2021” has to be changed by reference.
Altogether, all results are very original and interesting to the readers, and all cryo-EM maps should be made public to help the scientific community.
How do the authors suggest to validate these in silico data with in vitro experiments? It may be important to include this information in the conclusions.
Some studies reported that the number of trimers is related to different infectivity of HIV-1 strains. In which mode do you link this evidence to the structural model proposed in the present study?
Author Response
We are thankful to reviewer 2 for their kind evaluation of our manuscript and the careful feedback. Please find our comments below.
Reviewer 2
Despite the human immunodeficiency virus type 1 (HIV-1) was discovered in the early 1980s, significant knowledge gaps remain about the superstructure of the HIV-1 matrix (MA) shell. Understanding the HIV-1 structure and the interaction with Env glycoproteins is a key step for the HIV-1 entry process. The authors have already reported that Env proteins clustered in a space without MA shell, named “fusion hub”. In addition, there are different models of HIV-1 MA described in recent scientific literature, but in the present study, the authors propose an improved HIV-1 structural model. MA trimers are arranged in incomplete hexagons and lines, and Env proteins are packaged in the gap, which measures 66.3 nm ± 15.0 nm.
An extensive editing of English language and style are required (line 32, “Type 1 human immunodeficiency virus” is incorrect;
We modified the “Type 1 human immunodeficiency virus” to “Human immunodeficiency virus-1 (HIV-1)”
line 244, “we did identify” is incorrect).
The word “did” was deleted
Please, revise the reference style. It is not that recommended by the journal guidelines.
The reference style is now correct
Line 52, reference is missing; line 237, “HIV paper 2021” has to be changed by reference.
The reference was added
Altogether, all results are very original and interesting to the readers, and all cryo-EM maps should be made public to help the scientific community.
References in the Material and Methods to the cryo-EM maps are now included
Quantification of MA Shell Diameter and Gap: Cryogenic electron microscopy (cryo-EM) maps of HIV-1 particles, consisting of the phospholipid bilayer and matrix (MA) shells, were compiled from EMD-4020 and EMD-13085 from the electron microscopy database (https://www.ebi.ac.uk/emdb/).
How do the authors suggest to validate these in silico data with in vitro experiments? It may be important to include this information in the conclusions.
We’ve included a Conclusion. Researchers are encouraged to download the software available online and follow the step-by-step instructions. The new text in the conclusion is below.
In summary, we combine the knowledge accumulated on the structure of the MA sell and the Env trimers to create an updated model of the HIV-1 particle. We developed and made publicly available the software needed to build and test MA shell models in hopes that the community will avoid the use of hexagon tiles in their models. The size of the MA shell gap in which Env proteins likely interact was determined and thus we postulate that this region of the virus facilitates interactions between Env proteins via their CT domain. The bioinformatics software we have developed in this study will help us validate future cryo-EM data maps, a necessary step needed in data validation.
Some studies reported that the number of trimers is related to different infectivity of HIV-1 strains. In which mode do you link this evidence to the structural model proposed in the present study?
It has been suggested that the number of trimers increases infectivity. Although this is true, this manuscript does not attempt to answer the mechanism of entry and the interplay of trimer numbers in these dynamics.
Reviewer 3 Report
The Mangukia et al. Viruses manuscript, "Validation of HIV-1 MA Shell Structural Arrangements and Env Protein interactions Predict a Role of the MA Shell in Viral Maturation," presents a computer analysis of the HIV-1 matrix (MA) protein shell and how it may interact with the cytoplasmic tails (CTs) of the envelope (Env) protein. The manuscript is disappointing for its disregard for recently reported results in the field. Given that, the conclusions seem speculative. Specific concerns are as follows:
1. The authors have not fully taken into account the following results:
a. Qu et al., Science (2021): Although this is cited, the submitted manuscript does not fully address the appearance of immature and mature hexamer MA lattices demonstrated by cryo-EM in this paper.
b. Piai et al., Nature Communications (2020): The authors have ignored the trimeric CT baseplate structure reported in this paper, and instead base their models on an older paper (Murphy et al., 2017), which didn't yield CT trimers because the Env TM domain was not present.
c. Prasad et al., Cell (2022): The authors have ignored this paper, which showed how HIV-1 Env proteins are positioned relative to immature MA hexamers.
2. The authors have failed to include citations for the following statements:
a. Lines 96-98: "maps of HIV-1 particles...were compiled from PubMed." Note that the link given was inoperable, and citing ALL of PubMed is inexplicable.
b. Line 108: "models...were compiled from recent scientific literature." No citations were given.
c. Line 128: "maps from selected literature models." No citations were given.
d. Line 149: "Eight immature HIV particles." The authors did not indicate from where.
e. Line 165: "A representative Cryo-EM particle." From where?
f. Line 169: "A total of 8 HIV-1 particles were assessed." From where?
g. Line 177: "We located 5 papers" but failed to cite them here.
h. Line 237: The citation is "HIV paper 2021."
Author Response
Reviewer 2
Thank you reviewer 2 for your thoughtful analysis of our manuscript and the valuable feedback.
Comments and Suggestions for Authors
The Mangukia et al. Viruses manuscript, "Validation of HIV-1 MA Shell Structural Arrangements and Env Protein interactions Predict a Role of the MA Shell in Viral Maturation," presents a computer analysis of the HIV-1 matrix (MA) protein shell and how it may interact with the cytoplasmic tails (CTs) of the envelope (Env) protein. The manuscript is disappointing for its disregard for recently reported results in the field. Given that, the conclusions seem speculative. Specific concerns are as follows:
- The authors have not fully taken into account the following results:
- a) Qu et al., Science (2021): Although this is cited, the submitted manuscript does not fully address the appearance of immature and mature hexamer MA lattices demonstrated by cryo-EM in this paper.
Qu et al was a key manuscript influencing our work. We have made this clear by adding a paragraph in the discussion section.
A recent paper by Kun Qu et al [8], showed Cryo-EM tomography images of 3 adjacent hexagons. To our knowledge this work is the first published EM electron density map that shows direct evidence of hexagon-to-hexagon interactions. This evidence led us to update our MA shell model to include hexagons. Unfortunately, the map files were not available and only the images shown within the paper can be analyzed. Thus, we carefully extracted the triangle tile data from this manuscript and included this MA shell in our analysis.
- b) Piai et al., Nature Communications (2020): The authors have ignored the trimeric CT baseplate structure reported in this paper, and instead base their models on an older paper (Murphy et al., 2017), which didn't yield CT trimers because the Env TM domain was not present.
- c) Prasad et al., Cell (2022): The authors have ignored this paper, which showed how HIV-1 Env proteins are positioned relative to immature MA hexamers.
To address comments b and c we have added the following paragraph to the Discussion section. In short, we are unable to add the structures determined by cryo-EM by Prasad and the NMR structures by Piai because the CT domain of the Env proteins were significantly truncated and thus we are not convinced that their reported configurations are representative of the native interactions between the MA shell and the Env proteins.
The interaction of Env and the MA shell has been a focus by other researchers. Prasad et al [21] used a cryo-electron tomography of HIV-1 virus-like particles (VLPs) that display elevated levels of Env trimers on their surface. This is likely feasible due to a significant 102 amino acid deletion of the C-terminal CT domain of the gp41 protein. It is thus not surprising that the truncated gp41 can adopt 3D shapes that accommodate the Env trimer above the plane of the MA shell, as discovered in their electron tomography. We have previously described that the size of a full CT domain is predicted to occupy a 3D volume similar to that of an MA monomer [5]. Thus, it is unlikely that the interactions between the MA shell and Env trimers described by Prasad et al occur in WT HIV-1 particles. The limitations resulting from CT domain deletions are also observed in the NMR structures determined by Piai et al (PDB IDs: 6UJU and 6UJV) [22]. In their study, deletion of only 22 amino acids in the CT domain resulted in a plate-like arrangement of the remaining structure similar to that reported by Prasad et al. As a result, we were unable to use the NMR structures reported by Piai et al or the cryo-electron tomograms by Prasad et al.
- The authors have failed to include citations for the following statements:
- a) Lines 96-98: "maps of HIV-1 particles...were compiled from PubMed." Note that the link given was inoperable, and citing ALL of PubMed is inexplicable.
This link was fixed and the new sentence points to the correct files.
Cryogenic electron microscopy (cryo-EM) maps of HIV-1 particles, consisting of the phospholipid bilayer and matrix (MA) shells, were compiled from EMD-4020 and EMD-13085 from the electron microscopy database (https://www.ebi.ac.uk/emdb/).
- b) Line 108: "models...were compiled from recent scientific literature." No citations were given.
The correct references were added.
- c) Line 128: "maps from selected literature models." No citations were given.
The correct references were added.
- Line 149: "Eight immature HIV particles." The authors did not indicate from where.
The Materials and methods now explain in detail the data extraction and the reference to cryo-EM maps. We point the reader to the Materials and Methods by adding the text at the end of the sentence.
“Eight immature HIV particles were analyzed in PyMOL to obtain volume density information and trace the extent of the MA shell in 3D (Fig 2), see Materials and Methods.”
- e) Line 165: "A representative Cryo-EM particle." From where?
The reference was added.
“A representative Cryo-EM particle from EMD-4020 is shown with distances between same-colored pseudoatoms representing the matrix shell diameter and fusion gap length.”
- f) Line 169: "A total of 8 HIV-1 particles were assessed." From where?
The reference was added.
“HIV particle fusion hub measurements. A total of 8 HIV-1 particles were assessed from EMD-4020 and EMD-13085.”
- g) Line 177: "We located 4 papers" but failed to cite them here.
The references were added.
We conducted a literature search to locate published peer-reviewed articles that had used schematic representations of HIV-1 MA shells using hexagons or triangles. We located 4 papers and a total of 6 images were used as raw data for our validation [8-11]
- h) Line 237: The citation is "HIV paper 2021."
The reference was added.